# Enhanced Soil Fertility and Carbon Dynamics in Organic Farming Systems: The Role of Arbuscular Mycorrhizal Fungal Abundance

**DOI:** 10.3390/jof10090598

**Published:** 2024-08-24

**Authors:** So Hee Park, Bo Ram Kang, Jinsook Kim, Youngmi Lee, Hong Shik Nam, Tae Kwon Lee

**Affiliations:** 1Department of Environmental and Energy Engineering, Yonsei University, Wonju 26493, Republic of Korea; shpark0216@yonsei.ac.kr (S.H.P.); brkang@yonsei.ac.kr (B.R.K.);; 2Organic Agriculture Division, National Institute of Agricultural Sciences, Wanju 55365, Republic of Korea

**Keywords:** soil fertility, microbial diversity, agricultural practices, arbuscular mycorrhizal fungi, crops

## Abstract

Arbuscular mycorrhizal fungi (AMF) are critical for soil ecosystem services as they enhance plant growth and soil quality via nutrient cycling and carbon storage. Considering the growing emphasis on sustainable agricultural practices, this study investigated the effects of conventional and organic farming practices on AMF diversity, abundance, and ecological functions in maize, pepper, and potato-cultivated soils. Using next-generation sequencing and quantitative PCR, we assessed AMF diversity and abundance in addition to soil health indicators such as phosphorus content, total nitrogen, and soil organic carbon. Our findings revealed that, while no significant differences in soil physicochemical parameters or AMF diversity were observed across farming systems when all crop data were combined, organic farming significantly enhances AMF abundance and fosters beneficial microbial ecosystems. These ecosystems play vital roles in nutrient cycling and carbon storage, underscoring the importance of organic practices in promoting robust AMF communities that support ecosystem services. This study not only deepens our understanding of AMF's ecological roles but also highlights the potential of organic farming to leverage these benefits for improving sustainability in agricultural practices.

## 1. Introduction

Arbuscular mycorrhizal fungi (AMF) are fundamental biotic constituents of soil ecosystems that critically enhance plant nutrient uptake, soil structure, and carbon storage mechanisms [1]. AMF form symbiotic associations with over 80% of terrestrial plant species, facilitating phosphorus and nitrogen acquisition via hyphal growth, which is limited in soils but essential for plant growth [2]. AMF also contribute to soil aggregate formation, thereby improving soil aeration and water retention. They significantly influence carbon dynamics via increasing carbon allocation to plant roots and the biomass [3]. Approximately 5–20% of the plant total carbon uptake is transferred to AMF, which enhances soil organic carbon (SOC) accumulation below ground [4]. The functional roles of AMF in soil health underscore their potential in mitigating climate change impacts and enhancing agricultural sustainability, making them crucial for organic farming practices, wherein biotic interactions are relied on more than chemical inputs.

A number of studies have documented the impact of agricultural practices on AMF diversity and abundance, however, the results were inconsistent and vary from study to study, demonstrating the complexity of these interactions and the influence of localized environmental and management conditions. Intensive agricultural practices such as continuous monoculture cropping, high fertilizer and pesticide inputs, and intensive tillage generally reduce AMF diversity and abundance compared to low-input or natural systems [5,6,7]. In contrast, organic farming with low fertilizer inputs, diverse crop rotations, cover cropping, reduced tillage, and organic amendments tends to maintain a higher AMF diversity and abundance than farming with conventional high-input systems [8,9,10]. However, empirical data on the effects of these farming modalities on AMF diversity are inconsistent, necessitating comprehensive studies to examine these relationships across various cropping systems.

Exploring the specific interactions among AMF diversity, abundance, and soil health parameters is crucial for refining agricultural practices to maximize their ecological benefits. AMF diversity not only strengthens soil resilience against environmental challenges such as drought and nutrient scarcity via allowing different species to occupy unique niches and perform specialized roles but also enhances nutrient and carbon cycle stability [11,12]. High AMF concentrations extend the reach of hyphal networks beyond plant root zones, increasing nutrient absorption and reducing leaching, thereby boosting the efficiency of nutrient use within the soil [13]. AMF density is closely associated with increased soil organic matter via glomalin production, which helps stabilize carbon storage [14]. The detailed comparison of AMF colonization in chili pepper, potato, and maize reveals that, while common benefits such as enhanced growth and nutrient uptake are evident across these crops, significant differences in colonization intensity and response to nitrogen levels highlight the need for crop-specific management strategies [6]. For instance, chili pepper shows intensive colonization and specific gene expression patterns related to ammonium transporter genes which differ from those observed in potato and maize [15]. Understanding these dynamics will allow agricultural practices to be tailored to specific soil and crop needs, such as reducing phosphorus inputs where AMF efficiently mobilize phosphorus or synchronizing nitrogen availability with peak crop demands, which may potentially enhance yields under certain conditions without necessitating additional fertilizers. A holistic approach is needed to study the impact of changes in AMF diversity or abundance, as influenced by agricultural practices, on soil quality and ecosystem services.

This study aimed to elucidate the effects of conventional and organic farming practices on the diversity, abundance, and ecological functions of AMF in different crop types, including potato, pepper, and corn. Using next-generation sequencing (NGS) for diversity assessment and quantitative PCR (qPCR) to quantify AMF diversity and abundance, this study aimed to provide a detailed analysis of AMF dynamics across agricultural systems. We explored the correlations between AMF abundance and key soil health indicators, such as phosphorus content, total nitrogen, and soil organic carbon, to understand the potential contribution of AMF to nutrient cycling and carbon storage in agricultural soils. This comprehensive approach offers valuable insights into the role of AMF in enhancing soil ecosystem services, particularly under various agricultural management practices, thereby providing information on sustainable farming strategies.

## 2. Materials and Methods

### 2.1. Study Sites and Soil Sample Collection

The study area included 40 sampling sites across six distinct regions in South Korea (Appendix A). These sites were distributed between two farming systems (conventional and organic practices) and targeted three specific crops: chili peppers (*Capsicum annuum* var. *annuum*), potato (*Solanum tuberosum*), and maize (*Zea mays*). It is important to note that all samples were collected from open agricultural fields. For sites featuring organic farming, each was nationally certified to meet organic farming standards, ensuring the study's alignment with rigorous organic agricultural practices. The soil was sampled immediately prior to harvest for each crop in 2023 in the following order: potato from June to July, maize from June to September, and chili pepper from September to November. Soil samples were collected from two separate locations within each site, ensuring a minimum distance of 10 m between the sampling points. At each location, four rhizosphere subsamples were taken from a soil depth of up to 15 cm after the removal of the top 5 cm to minimize contamination from external sources. These subsamples were then combined, resulting in a total sample weight of approximately 500 g per location. The collected soil was immediately transported to the laboratory under refrigerated conditions to preserve its physicochemical and microbial integrity.

### 2.2. Soil Physicochemical Analyses

Soil pH and electrical conductivity (EC) was determined in a 1:5 soil/water suspension using a portable multiparameter meter (Orion Star™ A329, Thermo Fisher Scientific, Waltham, MA, USA). The SOC was determined using the K_2_Cr_2_O_7_-H_2_SO_4_ oxidation-reduction colorimetric method [16]. Total carbon (TC) and total nitrogen (TN) were analyzed using an elemental analyzer (Vario MICRO Cube; Elementar, Hesse, Germany). Available phosphorous (P) was extracted using the Lancaster method and detected at 720 nm using a UV/Vis spectrophotometer (Libra S80; Biochrom Ltd., Cambridge, UK). Exchangeable potassium, calcium, magnesium, and sodium were estimated using an inductively coupled plasma-optical emission spectrometer (iCAP PRO XP DUO, Thermo Fisher Scientific) after extraction with 1 M NH_4_OAc. The cation exchange capacity was evaluated via summing the exchangeable cation results.

### 2.3. Extraction of Glomalin-Related Soil Proteins

Glomalin-related soil proteins (GRSPs) were extracted via the methods established by Wright and Upadhyaya [17]. The process began with the extraction of easily extractable GRSPs (EEG), which involved autoclaving 1 g of soil with 8 mL of 20 mM sodium citrate, adjusted to a pH of 7.0 for 30 min at 121 °C. Totally extractable GRSPs (TEG) was obtained via autoclaving 1 g of soil in 8 mL of 50 mM sodium citrate buffer at 121 °C for 90 min. Extraction was repeated until the resulting supernatant was nearly clear. The samples were then immediately centrifuged at 5000× *g* for 15 min. The supernatant protein content was subsequently quantified using the Bradford protein assay kit (Bio-Rad Laboratories, Hercules, CA, USA). Each analysis was conducted in replicate.

### 2.4. DNA Extraction and qPCR

DNA was extracted from 0.5 g soil samples using a FastDNA SPIN kit for Soil (MP Biomedicals, Irvine, CA, USA) following the manufacturer’s instruction. The qPCR was performed in triplicate using a TB Green Premix (Takara Bio, Shiga, Japan) in 25 μL volume. To amplify bacterial 16S rRNA genes, primers 341F (5′-CCTACGGGNGGCWGCAG-3′) and 785R (5′-GACTACHVGGGTATCTAATCC-3′) [18] were used under the following PCR conditions: 95 °C for 4 min, 40 cycles of denaturation at 94 °C for 40 s, annealing at 52 °C for 30 s, and extension at 72 °C for 40 s. For total fungi, the primer sets FR1 (5′-AICCATTCAATCGGTAIT-3′) and FF390 (5′-CGATAACGAACGAGACCT-3′) [19] were used with the following PCR conditions: 95 °C for 8 min, 30 cycles of denaturation at 95 °C for 30 s, annealing at 50 °C for 45 s, and extension at 72 °C for 120 s with the same DNA template concentration as used for bacteria. AMF community quantification was based on the 18S rRNA gene fragment. The primers AMG1F (5′–ATAGGGATAGTTGGGGGCA T–3′) [20] and AM1 (5′-GTTTCCCGTAAGGCGCCGAA-3′) [21] were used with the following PCR conditions: 95 °C for 5 min, 35 cycles of denaturation at 95 °C for 15 s, annealing at 62 °C for 60 s, and extension at 72 °C for 20 s. Melting curve analysis was performed at the end of the qPCR cycle, starting at 60 °C and ending at 95 °C. The qPCR values were reported as the gene copy number g^−1^ dry soil. Each analysis was conducted in replicate.

### 2.5. High-Throughput Sequencing and Data Analysis

MiSeq libraries of the 18S rRNA region (small subunit ribosomal RNA) of AMF communities were prepared for each sample using primer set NS31 (5′-TTGGAGGGCAAGTCTGGTGCC-3′) [22] and AML2 (5′-GAACCCAAACACTTTGGTTTCC-3′) [23] to which overhang adapter sequences were attached. PCR assays were performed using the Herculase fusion II DNA polymerase dNTP combo (Agilent Technologies, Santa Clara, CA, USA). The first PCR was conducted with the following PCR conditions: 2 min at 98 °C, followed by 35 cycles of 10 s denaturation at 98 °C, 30 s annealing at 56 °C, 30 s of extension at 72 °C, and a final elongation for 2 min at 72 °C. Sequencing libraries were generated using the Nextera XT Index Kit v2 (Illumina Inc., San Diego, CA, USA) with the application of unique dual Nextera indices according to the manufacturer’s protocols. Libraries were cleaned with AMPure XP beads (Beckman Coulter, Brea, CA, USA) and the enriched library quality was evaluated using a microplate reader (Spark 10M, Tecan, Zürich, Switzerland). The libraries were subsequently sequenced on the Illumina MiSeq platform (Macrogen, Seoul, Republic of Korea). Each analysis was conducted in replicate.

The raw paired sequences were assembled, demultiplexed, and filtered using Quantitative Insights into Microbial Ecology (QIIME2 v2020.11) [24]. Operational taxonomic units (OTUs) were clustered using sequences from all 40 AMF community libraries at a 97% identity level. Taxonomic assignments were completed using the MaarjAM database [25]. The sequences were deposited in the NCBI Sequence Read Archive under accession number PRJNA1125147.

### 2.6. Statistical Analyses

All statistical analyses were performed using R software (ver. 4.4.1) [26]. The Shapiro–Wilk normality test was employed to assess data distribution normality. The alpha diversity (richness) of the AMF community within each sample was quantified using the R package ‘vegan’ (ver. 2.6-4) [27]. Non-metric multidimensional scaling (NMDS) analysis was performed to compare the beta diversity of the AMF community utilizing the Bray–Curtis dissimilarity index in ‘vegan’. The significant differences among soil properties, microbial diversity and abundance, and glomalin production between conventional and organic farming was analyzed using a Wilcoxon’s rank-sum test. Spearman’s rank correlation coefficients were determined through the soil properties and microbial abundance, and the AMF diversity was determined through agricultural practices.

## 3. Results

### 3.1. No Impact of Agricultural Practice on Soil Physicochemical Parameters

We conducted physicochemical analyses on soils cultivated with maize, pepper, and potato under conventional and organic agricultural practices. The results indicated significant differences in a few soil parameters between conventional and organic farming systems for individual crops (Table 1). However, when data from all three crops were combined, no significant differences were observed in soil parameters between the two farming systems. These findings suggest that, while organic and conventional practices distinctly influence soil physicochemistry on a crop-specific basis, these effects are not consistently observable across different crop types.

### 3.2. AMF Diversity Unaffected by Farming Practices across Crops

We evaluated the alpha and beta diversity of AMF across different agricultural practices and crop types. The results revealed that alpha diversity was significantly lower in pepper when grown under organic practices compared to conventionally cultivated pepper; however, no significant differences were observed in maize, potato, or when all crops were collectively considered (Figure 1a). Beta diversity was analyzed via NMDS, which indicated no significant differences attributable to agricultural practices (ANOSIM, R = 0.006, *p*-value = 0.397), suggesting that the agricultural practice type did not significantly alter AMF community composition (Figure 1b). However, significant statistical differences were observed in beta diversity among the different crops (ANOSIM, R = 0.150, *p*-value < 0.05), though these differences did not show a clear pattern related to specific crops, as depicted in the NMDS plots (Figure 1c). This suggests that, while crop type influences AMF community structures, the specific agricultural practice does not markedly affect the overall AMF diversity.

### 3.3. Stable Microbial Abundances Revealed via qPCR in Different Farming Practices across Crops

Using qPCR, we analyzed the abundance of bacteria (16S rRNA genes), fungi (18S rRNA genes), and AMF (18S rRNA genes) in soils cultivated with maize, pepper, and potato under different agricultural practices. Our results showed no statistically significant differences in the gene copy numbers of the three genes among the crops or between conventional and organic farming practices (Figure 2). Specifically, the 16S gene copy numbers ranged from approximately 5.4 × 10^8^ to 3.3 × 10^10^ copies/g soil, with no significant difference across the farming practices (Figure 2a). Fugal 18S gene copy numbers fluctuated from approximately 1.2 × 10^8^ to 2.1 × 10^11^ copies/g soil across all crops and practices (Figure 2b). For AMF, gene copy numbers consistently hovered around 1.2 × 10^6^ to 2.1 × 10^8^ copies/g soil regardless of the crop or farming practice (Figure 2c). Overall, these results indicated that the overall abundance of AMF, fungi, and bacteria remained comparable between conventional and organic practices across the different crops. This suggests that agricultural practice type does not significantly affect the microbial abundance in the agricultural soil environments under study.

The qPCR analysis evaluated the relative AMF proportions to bacteria and total fungi across different crops cultivated under conventional and organic farming practices. For the AMF to bacteria ratio, organic practices generally exhibited an average of 1.0 (±0.4%) across maize, pepper, and potato, which was slightly higher than the range observed in conventional practices (0.8 ± 0.3%) (Figure 3a). In maize, the median percentage of AMF gene copies under the conventional practice was slightly lower compared to that in organic practices, though it was not statistically significant. Pepper and potato crops exhibited a similar trend, with organic farming showing slightly higher or comparable median values to conventional practices, suggesting a more favorable environment for AMF in organic farming systems across different crop types. The AMF to total fungi ratio showed higher variation among each sample compared to bacteria (1.1 ± 1.0% from conventional practices and 1.5 ± 1.1% from organic practices) (Figure 3b). These results indicate that the abundance of AMF gene copies varies minimally between conventional and organic practices across different crops, showing slight differences but no substantial shifts.

### 3.4. AMF Abundance Correlates with Soil Nutrient and Carbon in Organic Farming Practices

We analyzed the correlation between the abundance of bacteria, fungi, and AMF with nutrient and soil carbon indices across different agricultural practices. The results revealed distinct correlation patterns dependent on the type of microorganism and agricultural practice (Figure 4). Bacteria showed a statistically significant positive correlation with SOC, TEG, TN, and phosphate (P_2_O_5_) levels in organic farming systems and TC, TEG, and TN in conventional practices (Figure 4a). In organic farming in particular, correlation coefficients with P_2_O_5_ and SOC were three times higher compared to those in conventional farming. Fungi exhibited less significance with soil properties (Figure 4b). Fungal abundance was significantly correlated with TEG and P_2_O_5_ in conventional practices and with TN in organic practices. AMF demonstrated a similar pattern with bacterial abundance (Figure 4c). AMF showed robust significant positive correlations with all studied soil parameters across farming practices. Particularly, the correlation coefficients with SOC and P_2_O_5_ were 2 and 4.2 times higher in organic farming, respectively, than those in conventional farming. A statistically significant and positive correlation in the extraction of easily extractable GRSPs was only found for organic farming. This pattern underscores the enhanced role of AMF in nutrient cycling and soil carbon dynamics in organically managed soils, suggesting a pivotal contribution to soil health and fertility via microbial activity.

## 4. Discussion

This study elucidates the complex relationships among agricultural practices, AMF diversity and abundance, and their consequent impacts on soil health and ecosystem services. Our findings reveal that, although the physicochemical characteristics and AMF diversity and abundance remain relatively consistent across various crops and agricultural practices, AMF significantly impact soil health primarily in fields managed organically. This effect is particularly notable in the enhanced availability of nutrients such as nitrogen and phosphorus, as well as in soil carbon accumulation, specifically in SOC and TEG. These results suggest employing organic practices that reduce soil disturbance and maintain plant diversity, as these elements are more effective at sustaining abundant and diverse AMF communities that optimize these beneficial effects. This will maximize the positive effects on soil health.

Our results across crops revealed no significant differences in soil physicochemical parameters when comparing conventional and organic farming systems. This result suggests that the influence of agricultural practices on soil properties is highly crop-specific and may not be generalized across diverse cropping systems. The variability observed in individual crop analyses suggests that certain crops may respond uniquely to similar agricultural practices. For example, for the pepper crop, organic practices may increase soil nitrogen levels because of better integration of organic fertilizers and enhanced microbial activity that facilitates nitrogen fixation [28]. Conversely, for the maize crop, the high nutrient demand may overshadow the subtle benefits conferred by organic practices, causing negligible differences in soil parameters compared to conventional methods [29]. While individual crop analyses did exhibit some variability, the lack of consistent trends across all crops implies that broader environmental and management factors may mitigate or obscure the effects of specific agricultural practices.

The observed lack of significant differences in AMF alpha and beta diversity between conventional and organic farming systems aligns with previous research indicating the resilience of AMF communities to various agricultural practices. While organic farming practices are generally presumed to enhance microbial diversity because of reduced chemical inputs and increased organic amendments [30], our study suggests that AMF diversity may not be as sensitive to these changes as initially hypothesized. Some studies found no significant differences in AMF diversity between organic and conventional systems for certain crops [31,32]. This could be because of factors such as past management history, soil nutrient levels, climate, plant diversity, or crop-specific effects overshadowing farming system impacts [33,34]. The significant reduction in alpha diversity observed in pepper-cultivated soil under organic practices could be attributed to crop-specific interactions or localized soil conditions that warrant further investigation. This underscores the importance of considering various elements for agricultural practices in managing AMF populations to enhance soil health.

Our findings demonstrate strong positive correlations between AMF abundance and key soil health indicators such as phosphorus content and SOC under organic farming practices. AMF form beneficial symbiotic relationships with plant roots, facilitating the exchange of nutrients essential for plant growth; for example, AMF hyphae extend into the soil, accessing phosphorus and nitrogen beyond the immediate reach of plant roots, thereby improving plant nutrient absorption efficiency [35]. This symbiotic relationship is especially advantageous in organic systems wherein reliance on synthetic fertilizers is reduced and plants depend more on natural biological processes for nutrient acquisition [36]. Organic farming practices such as reduced tillage help preserve AMF networks and avoid physical disruptions caused by conventional tillage that impair AMF colonization [37]. Moreover, organic farms avoid the use of AMF-harmful fungicides and pesticides and maintain lower soil phosphorus levels, encouraging plants to utilize AMF for nutrient uptake [38,39,40]. Increasing crop diversity and applying organic amendments in organic systems promote richer and more diverse AMF communities, contrasting with conventional agriculture, which often employs modern crop varieties that are less conducive to AMF symbiosis [41]. These practices in organic farming enhance the role of AMF, allowing them to thrive and significantly contribute to sustainable agricultural ecosystems.

The observed similarities in the patterns of bacterial and AMF abundances, particularly in relation to SOC and P_2_O_5_, reflect a synergistic interaction in organically managed systems that significantly influences soil nutrient dynamics. Bacteria and AMF both play crucial roles in the biogeochemical processes that mediate nutrient availability and cycling within the soil [42]. Bacteria are involved in organic matter decomposition, nitrogen fixation, and phosphate solubilization, whereas AMF primarily enhance the uptake of phosphorus and other micronutrients via extensive hyphal networks that effectively increase the root surface area in contact with soil. Furthermore, the observation that the ratio of AMF to bacteria is slightly higher in organic farming suggests a nuanced shift in microbial community dynamics where AMF may be relatively more influential in these systems compared to conventional practices. This could be indicative of a soil environment in organic systems that favors AMF growth or activity, possibly because of higher organic matter content, which AMF are adept at utilizing. This shift is significant, as it may enhance the ability of crops to access soil nutrients that are less mobile and harder to acquire, such as phosphorus, through the symbiotic relationships that AMF establish with plant roots [43]. These interrelated dynamics highlight the complexity of microbial interactions in soil and underscore the potential of organic farming practices to harness these interactions for sustainable agriculture. Organic practices may enhance the overall efficiency of nutrient uptake by plants via promoting a higher ratio of AMF to bacteria, hence contributing to a cycle of soil health improvement and sustainable yield enhancement.

Despite the comprehensive insights provided by this study regarding the relationships between agricultural practices, AMF diversity, and soil health, it has several limitations. Although the resilience of AMF diversity towards conventional versus organic practices found in this work is consistent with some prior studies, further research into the complex interactions between AMF and different agricultural inputs is required, particularly in regard to varying climates, past management history, and soil conditions. This study did not concurrently analyze the diversity and structure of bacterial communities nor their beta diversity, which limits our understanding of complete microbial dynamics and their interactions within the soil. Including bacterial diversity and structure analyses in future research could provide a more holistic view of the microbial ecosystem and its response to agricultural practices. Future research should focus on long-term studies that incorporate a broader range of environmental variables and more detailed tracking of microbial dynamics over time. This would facilitate understanding the long-term impacts of different farming practices on AMF communities and their functional roles in ecosystem services. Moreover, expanding the scope to include more diverse crop systems and integrating advanced molecular techniques could expand existing knowledge on the specific mechanisms of AMF contributions to soil health and plant productivity under varying agricultural regimes.

## 5. Conclusions

Our study provides comprehensive insights into the dynamics of AMF in conventional and organic farming systems and elucidates their critical roles in nutrient cycling and soil carbon storage. Organically managed fields in particular demonstrate significant benefits in terms of AMF activity, which correlates strongly with improved soil nutrient content and enhanced carbon storage. This differentiation in the impact of AMF between farming systems necessitates developing agricultural practices that support and leverage the ecological roles of AMF, particularly in organic systems wherein they are integral to achieving sustainability and enhancing soil function. The implications of this study highlight the benefit of organic farming in enhancing soil fertility and carbon storage, thereby offering valuable in-sights for developing sustainable agricultural strategies. Further research should aim to dissect these complex interactions over longer periods and across more diverse environmental conditions, employing advanced molecular techniques to refine our understanding of the tailoring of sustainable farming practices to maximize ecological benefits and crop productivity. Future agricultural management strategies should consider fostering AMF and other beneficial soil microbes to sustainably maintain ecosystem services and improve crop production.

## Figures and Tables

**Figure 1 jof-10-00598-f001:**
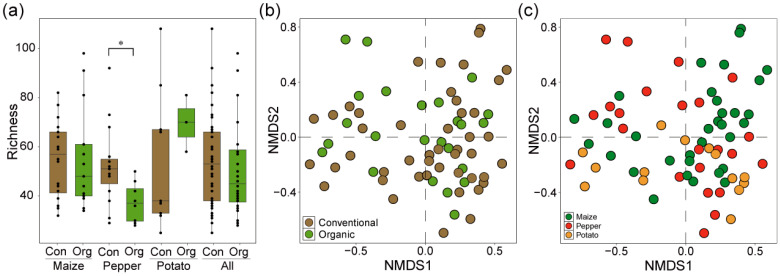
Assessment of arbuscular mycorrhizal fungi (AMF) diversity in conventional and organic agricultural systems across different crops. (**a**) Alpha diversity (richness) of AMF in maize, pepper, and potato cultivated under conventional (Con) and organic (Org) farming practices. Significant differences between conventional and organic systems are denoted by an asterisk (*). Non-metric multidimensional scaling (NMDS) analysis of AMF beta diversity categorized by agricultural practices (**b**) and crop types (**c**) with a stress value of 0.229.

**Figure 2 jof-10-00598-f002:**
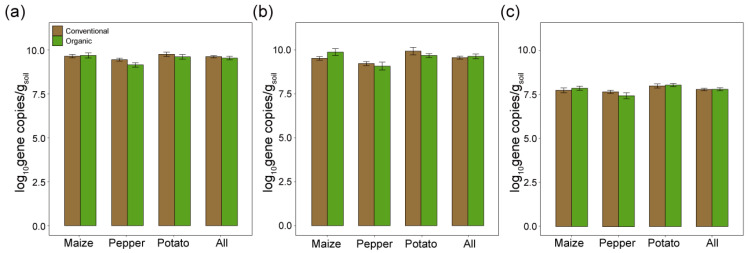
Quantitative PCR analysis of microbial abundance in agricultural soils. Log-transformed gene copy numbers per gram of soil for bacteria (**a**), fungi (**b**), and arbuscular mycorrhizal fungi (AMF) (**c**) across crops and combined samples in conventional and organic farming systems. Error bars indicate standard deviations from the mean.

**Figure 3 jof-10-00598-f003:**
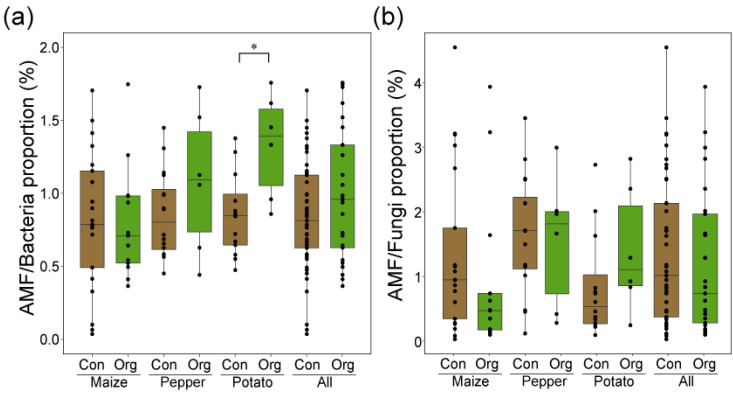
Relative proportions of arbuscular mycorrhizal fungi (AMF) to bacteria and fungi in conventional and organic agricultural systems across different crops. (**a**) The proportion of AMF to bacteria under conventional and organic farming practices across crops and combined samples. (**b**) The proportion of AMF to fungi for the same set of crops and agricultural practices. Asterisks (*) indicate statistically significant correlations (*p* < 0.05).

**Figure 4 jof-10-00598-f004:**
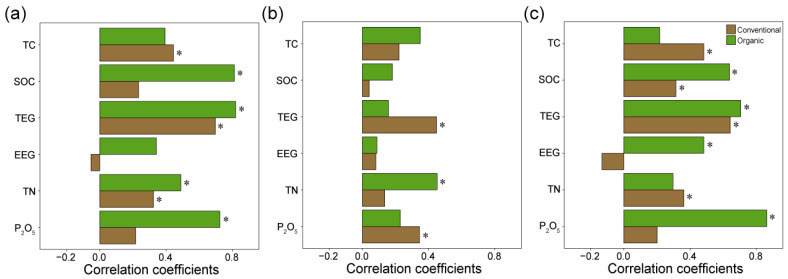
Correlation coefficients between microbial abundance, soil nutrients, and carbon storage indices. (**a**) Spearman’s rank correlation coefficients between bacterial abundance and various soil parameters under conventional and organic farming practices. Parameters include total carbon (TC), soil organic carbon (SOC), total extractable glomalin-related soil proteins (GRSPs) (TEG), easily extractable GRSPs (EEG), total nitrogen (TN), and available phosphorus (P_2_O_5_). Correlations for total fungi (**b**) and arbuscular mycorrhizal fungi (AMF) (**c**) with the soil parameters. Asterisks (*) indicate statistically significant correlations (*p* < 0.05).

**Table 1 jof-10-00598-t001:** Comparison of soil physicochemical parameters in conventional and organic farming systems across different crops. Values are mean ± standard deviation. Statistical significance (*p* < 0.05) between conventional and organic systems is indicated by asterisks (*).

	Maize		Pepper		Potato		All	
	Conventional (*n* = 10)	Organic (*n* = 7)	Conventional (*n* = 8)	Organic (*n* = 4)	Conventional (*n* = 8)	Organic (*n* = 3)	Conventional (*n* = 26)	Organic (*n* = 14)

pH	6.0 ± 0.6	5.9 ± 0.9	6.5 ± 0.6	6.7 ± 0.7	6.4 ± 0.9	6.7 ± 0.8	6.2 ± 1.0	6.3 ± 0.9
EC (mS/cm)	264.4 ± 250.7	305.1 ± 120.1	302.5 ± 251.2	123.4 ± 71.2 *	461.3 ± 265.7	216.9 ± 170.3	323.9 ± 262.5	232.8 ± 138.5
SOC (g/kg)	13.2 ± 5.5	13.5 ± 4.0	17.5 ± 11.4	8.4 ± 5.2 *	13.8 ± 3.1	15.2 ± 13.7	14.7 ± 7.9	12.2 ± 6.8
TC (%)	2.1 ± 1.8	1.9 ± 0.5	2.1 ± 1.1	1.1 ± 0.8 *	1.5 ± 0.9	1.0 ± 0.7	1.9 ± 1.4	1.5 ± 0.7
TN (%)	0.2 ± 0.2	0.2 ± 0.0	0.2 ± 0.1	0.1 ± 0.1 *	0.2 ± 0.1	0.1 ± 0.1	0.2 ± 0.2	0.2 ± 0.1
C:N ratio	8.6 ± 1.3	9.3 ± 0.9	9.0 ± 1.5	7.7 ± 3.6	15.7 ± 30.5	6.8 ± 4.7	10.4 ± 14.6	8.4 ± 2.8
P_2_O_5_ (mg/kg)	495.1 ± 349.2	786.8 ± 351.3 *	878.9 ± 281.1	421.8 ± 179 *	1208.1 ± 356	1014.2 ± 605.6	790.8 ± 444.3	706.4 ± 403.9
EEG (mg_protein_/g_soil_)	3.9 ± 1.6	3.1 ± 1.0	3.1 ± 1.3	2.0 ± 1.4 *	4.4 ± 1.2	4.0 ± 0.8	3.8 ± 1.5	3.0 ± 1.3
TEG (mg_protein_/g_soil_)	7.2 ± 1.9	8.4 ± 4.2	8.3 ± 4.0	4.6 ± 1.9 *	9.7 ± 3.9	8.1 ± 3.3	8.3 ± 3.4	7.2 ± 3.8 *
Ca (cmol/kg)	4.6 ± 1.9	5.8 ± 2.0	6.9 ± 2.3	5.3 ± 2.2	5.7 ± 1.5	5.9 ± 4.1	5.6 ± 2.3	5.6 ± 2.4
K (cmol/kg)	0.7 ± 0.8	0.9 ± 0.7	0.5 ± 0.4	0.2 ± 0.1 *	1.8 ± 1.2	0.9 ± 1.3	0.9 ± 1.0	0.7 ± 0.8
Mg (cmol/kg)	1.3 ± 0.8	2.0 ± 0.6 *	2.2 ± 1.2	2.1 ± 1.5	2.0 ± 0.7	1.9 ± 1.5	1.8 ± 1.0	2.0 ± 1.1
Na (cmol/kg)	0.2 ± 0.2	0.2 ± 0.1 *	0.2 ± 0.1	0.3 ± 0.4	0.3 ± 0.3	0.2 ± 0.3	0.2 ± 0.2	0.2 ± 0.2
CEC (cmol/kg)	9.2 ± 2.7	12.1 ± 2.2 *	11.5 ± 3.5	8.8 ± 3.5	11.9 ± 4.1	9.9 ± 6.4	10.6 ± 3.7	10.7 ± 3.7

EC: electrical conductivity; SOC: soil organic carbon; TC: total carbon; TN: total nitrogen; EEG: easily extractable GRSP; TEG: totally extractible GRSP; CEC: cation exchange capacity.

## Data Availability

The original contributions presented in the study are included in the article/Appendix A, further inquiries can be directed to the corresponding authors.

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
