# Peer review of "Enhanced Soil Fertility and Carbon Dynamics in Organic Farming Systems: The Role of Arbuscular Mycorrhizal Fungal Abundance"

_jof, 2024, doi:10.3390/jof10090598_

Round 1

Reviewer 1 Report

The paper entitled “Enhanced soil fertility and carbon dynamics in organic farming systems: The role of arbuscular mycorrhizal fungal abundance” by Park et al. was set out to elucidate the effects of conventional and organic farming prac-67 tices on the diversity, abundance, and ecological functions of AMF in different crop types. Some of the results are scientifically interesting: organic practices support a more robust AMF community. However the whole study is in fact very descriptive. The papers could not demonstrate a enough level of novelty and originality for considering publication in Journal of Fungi. Unfortunately, I do not recommend this manuscript in the present state for publication in the journal.

Specific comments

1.      Abstract The significance of the present study is not clear.

2.      Line 42: The following description “Despite the recognized benefits of AMF, the influence of agricultural practices on 42 their diversity and abundance remains unclear with conflicting outcomes”is not true.

3.      Line 79: The methods of soil sample collection is vague. Both the procedure of collection of rhizosphere soil samples and why the three crops were collected as specific crops are not clear.

4.      The status of AM colonization of the three crops needs give more information.

5. The manuscript requires substantial grammatical revisions in its present form and needs a big improvement.

Author Response

The paper entitled “Enhanced soil fertility and carbon dynamics in organic farming systems: The role of arbuscular mycorrhizal fungal abundance” by Park et al. was set out to elucidate the effects of conventional and organic farming practices on the diversity, abundance, and ecological functions of AMF in different crop types. Some of the results are scientifically interesting: organic practices support a more robust AMF community. However, the whole study is in fact very descriptive. The papers could not demonstrate an enough level of novelty and originality for considering publication in Journal of Fungi. Unfortunately, I do not recommend this manuscript in the present state for publication in the journal.

Specific comments

Comment 1 : Abstract The significance of the present study is not clear.

Response 1 : We thank you for your feedback regarding the clarity of the significance of our study. We have revised the abstract for clarity [L16-23].

Comment 2 : Line 42: The following description “Despite the recognized benefits of AMF, the influence of agricultural practices on their diversity and abundance remain unclear with conflicting outcomes” is not true.

Response 2 : We thank you for pointing out the potential misunderstanding in our statement regarding the impact of agricultural practices on AMF diversity and abundance. We meant that numerous studies have indeed explored this area and documented various effects. Our intention was to highlight appearing inconsistent or contradictory in previous studies due to differences in methodology, geographic focus, crop types, and specific farming practices evaluated. To clarify our position, we have revised the statement in our manuscript to better reflect the current state of research [L40-43].

Comment 3 : Line 79: The methods of soil sample collection is vague. Both the procedure of collection of rhizosphere soil samples and why the three crops were collected as specific crops are not clear. The status of AM colonization of the three crops needs give more information.

Response 3 : We thank the reviewer for your valuable feedback requesting further details on the status of AMF colonization in chili pepper, potato, and maize. We agree that a deeper understanding of these interactions is crucial for interpreting our results. We have added our manuscript to include a concise comparison of AM fungi colonization among the three crops, emphasizing both the commonalities and distinctions in their interactions with AM fungi in Introduction section [L60-66].

Comment 4 : The manuscript requires substantial grammatical revisions in its present form and needs a big improvement.

Response 4 : Prior to submission, our manuscript underwent thorough English proofreading by two professional editors from a recognized English proofreading organization. We will include the certificates of proofreading with our resubmission as evidence of this professional review. We believe that these efforts have significantly enhanced the manuscript's language quality.

Reviewer 2 Report

A valuable manuscript with good presentation of material and methods, results and comprehensive conclusions.

Line 63: The statement 'synchronizing nitrogen availability with peak crop demands to enhance 63 yields without extra fertilizers' is very bold and should be formulated more cautiously.

Table 1: Does +/- in the table represent the standard deviation or the standard error? Please add.

Figure 1, (b) and (c) appear redundant to the reviewer and should be omitted. This of course also applies to the mention in the text.

Author Response

A valuable manuscript with good presentation of material and methods, results and comprehensive conclusions.

Comment 1 : Line 63: The statement 'synchronizing nitrogen availability with peak crop demands to enhance yields without extra fertilizers' is very bold and should be formulated more cautiously.

Response 1 : We have revised this statement to reflect the potential benefits while clearly indicating that these outcomes are contingent on specific agronomic and environmental conditions [L69-70].

Comment 2 : Table 1: Does +/- in the table represent the standard deviation or the standard error? Please add.

Response 2 : We thank the reviewer for highlighting the ambiguity regarding the statistical metrics used in Table 1. To clarify, the +/- values originally represented the standard deviation, not the standard error. We have now added Table 1 to state this [L185-186].

Comment 3 : Figure 1, (b) and (c) appear redundant to the reviewer and should be omitted. This of course also applies to the mention in the text.

Response 3 : We understand the reviewer's concern about the apparent similarity between these two panels. However, each panel serves a distinct purpose by illustrating different aspects of our data. Panel (b) is color-coded to emphasize the effects of agricultural practices, while panel (c) uses a different color scheme to highlight the impacts associated with different crop types. This dual presentation allows for a clearer visual comparison of how each factor—agricultural practices and crop types—independently influences AMF community composition.

Reviewer 3 Report

Suppose I compare existing knowledge within the manuscript and published works. In that case, most of the research is based on Arbuscular Mycorrhizal Fungi, which reinforces the established understanding of their crucial role in enhancing plant growth and soil quality through nutrient cycling and carbon storage. The evaluated study is focused on the impact of conventional and organic farming practices on Arbuscular Mycorrhizal Fungi diversity and abundance, which aligns with ongoing efforts to promote sustainable agriculture.

In the manuscript, there are new main points:

The study compares conventional and organic farming, highlighting the benefits of organic practices on Arbuscular Mycorrhizal Funghi communities.

Modern sequencing and quantitative PCR methods were used to obtain the results.

The crop-specific effect is identified depending on farming practices.

Evaluated also correlated with soil health, which is important for soil fertility and carbon dynamics.

The benefits of organic farming are also mentioned.

The study (manuscript) generally compares farming practices and uses advanced genomic tools to offer detailed insights into Arbuscular Mycorrhizal Funghi ecological functions. This point of view ridges gaps in the literature by highlighting crop-specific management strategies and empirically validating the benefits of organic farming for microbial and soil health.

The abstract is consistent with the text and informative, covering all the authors' main findings.
The introduction part is relatively short, but it contains everything needed. Both information on the importance of Arbuscular mycorrhizal fungi and the reasons why it is necessary to research them, as well as information on the two selected cultivation systems.
The aims of the work are defined and logical.
Material and methods - please provide more information directly in the text (considering the number of sampled locations), at least as a summary of what conditions were involved - probably specific to the crops - whether open field or greenhouse production.
The results chapter is informative and logically organized - there are tables with data and illustrative graphs. There is no need to make any adjustments here.
Discussion is a specific summary (generalization) of the results found. Information is confronted with relevant scientific works.
Conclusions - consistent, informative. Brief information on the achieved results, their applicability in practice, and directions for future research are outlined.

Author Response

Suppose I compare existing knowledge within the manuscript and published works. In that case, most of the research is based on Arbuscular Mycorrhizal Fungi, which reinforces the established understanding of their crucial role in enhancing plant growth and soil quality through nutrient cycling and carbon storage. The evaluated study is focused on the impact of conventional and organic farming practices on Arbuscular Mycorrhizal Fungi diversity and abundance, which aligns with ongoing efforts to promote sustainable agriculture. In the manuscript, there are new main points:

The study compares conventional and organic farming, highlighting the benefits of organic practices on Arbuscular Mycorrhizal Funghi communities.

Modern sequencing and quantitative PCR methods were used to obtain the results.

The crop-specific effect is identified depending on farming practices.

Evaluated also correlated with soil health, which is important for soil fertility and carbon dynamics.

The benefits of organic farming are also mentioned.

The study (manuscript) generally compares farming practices and uses advanced genomic tools to offer detailed insights into Arbuscular Mycorrhizal Funghi ecological functions. This point of view ridges gaps in the literature by highlighting crop-specific management strategies and empirically validating the benefits of organic farming for microbial and soil health.

Commemt 1 : The abstract is consistent with the text and informative, covering all the authors' main findings.
The introduction part is relatively short, but it contains everything needed. Both information on the importance of Arbuscular mycorrhizal fungi and the reasons why it is necessary to research them, as well as information on the two selected cultivation systems. The aims of the work are defined and logical.

Response 1 : We greatly appreciate the reviewer's in-depth analysis of our manuscript and their recognition of its contributions to the field of sustainable agriculture. We are pleased that the reviewer acknowledges the alignment of our work with existing knowledge while also appreciating our efforts to advance understanding through specific, novel insights.

Comment 2 : Material and methods - please provide more information directly in the text (considering the number of sampled locations), at least as a summary of what conditions were involved - probably specific to the crops - whether open field or greenhouse production.

Response 2 :We thank you for your suggestion to enhance the clarity of the Materials and Methods section of our manuscript. Detailed information regarding the number and specific locations of our samples is comprehensively documented in Table S1. We have revised the Materials and Methods section to state that all samples were collected from open field agricultural fields. Additionally, we emphasized that in the case of organic farming, all samples originated from nationally certified organic farms, ensuring that the study conditions meet strict organic agricultural standards. We believe that this additional detail directly in the text will help clarify the conditions under which the data was collected, providing better insight into the environmental variables that could influence the results [L89-92, L94-99].

Comment 3 : The results chapter is informative and logically organized - there are tables with data and illustrative graphs. There is no need to make any adjustments here.’

Discussion is a specific summary (generalization) of the results found. Information is confronted with relevant scientific works.

Conclusions - consistent, informative. Brief information on the achieved results, their applicability in practice, and directions for future research are outlined.

Response 3 : We sincerely appreciate your positive feedback on the organization and content of the Results, Discussion, and Conclusions sections of our manuscript.

Round 2

Reviewer 1 Report

Overall, the authors have improved the manuscript according to my comments. The revised manuscript has been significantly improved.

Overall, the authors have improved the manuscript according to my comments. The revised manuscript has been significantly improved.